# Isothermal Diffusion Behavior and Surface Performance of Cu/Ni Coating on TC4 Alloy

**DOI:** 10.3390/ma12233884

**Published:** 2019-11-24

**Authors:** Nan Wang, Yong-Nan Chen, Long Zhang, Yao Li, Shuang-Shuang Liu, Hai-Fei Zhan, Li-Xia Zhu, Shi-Dong Zhu, Yong-Qing Zhao

**Affiliations:** 1School of Materials Science and Engineering, Chang’an University, Xi’an 710064, China; wangnanchd@163.com (N.W.); zhanglongsir@163.com (L.Z.); wjsliyao@163.com (Y.L.); Sh_shuang@163.com (S.-S.L.); 2School of Chemistry, Physics and Mechanical Engineering, Queensland University of Technology (QUT), Brisbane, QLD 4001, Australia; 3CNPC Tubular Goods Research Institute, Xi’an 710017, China; zhulx@cnpc.com.cn; 4School of Materials Science and Engineering, Xi’an Shiyou University, Xi’an 710065, China; 5Northwest Institute for Nonferrous Metal Research, Xi’an 710016, China; trc@c-nin.com

**Keywords:** titanium alloys, isothermal diffusion, intermetallic compounds, microvoids, surface performance

## Abstract

The poor surface performance of titanium alloys substantially limits their application in many fields, such as the petrochemical industry. To overcome this weakness, the Cu and Ni double layers were deposited on the surface of TC4 alloy by the electroplating method, and the isothermal diffusion process was performed at 700 °C to enhance the binding ability between Cu and Ni layers. The isothermal diffusion behavior and microstructure of the coating were systematically analyzed, and tribological property and corrosion resistance of the coating were also evaluated to reveal the influence of isothermal diffusion on the surface performance. It was shown that multiple diffusion layers appeared on the Cu/Ni and Ni/Ti interface, and that Ni_x_Ti_y_ and Cu_x_Ti_y_ phases were formed in the coating with the increase of diffusion time. More importantly, Kirkendall diffusion occurred when the diffusion time increased, which led to the formation of continuous microvoids and cracks in the diffusion layer, weakening the surface performance of the Cu/Ni coatings. This paper unveils the relationship between the microstructure of the Cu/Ni coatings and isothermal diffusion behavior, providing guidelines in preparing high performance surface coatings.

## 1. Introduction

TC4 alloys have been broad utilized in petrochemical industry for their excellent corrosion resistance and high specific strength [1,2,3]. However, the poor thermal conductivity and wear resistance substantially restrict their large-scale application in oil and gas fields, especially as pipe coupling [4,5]. For this reason, many surface treatments of titanium alloys, such as micro-arc oxidation [6], anodic oxidation [7], and electroplating [8,9], have been developed to overcome these limitations and prolong service life. Among them, the thermal diffusion process between dissimilar materials is believed to be an effective approach in improving the surface performance of coating. For instance, the Cu coating on TC4 alloy by diffusion process can vastly strengthen surface hardness and wear resistance [10]. Furthermore, it has been proven that the introduction of suitable interlayer between Cu coating and TC4 alloy substrate by diffusion process can remarkably enhance its performance [11].

As is well known, the Kirkendall effect occurs during the process of atomic diffusion, and Tavoosi [12] believes that the numerous continuous microvoids or cracks are formed, making the microstructure of the diffusion layer imperfect and affecting the surface properties. However, Chen et al. [13] revealed that the micro-hardness was improved during the process of diffusion due to the formation of intermetallic compounds. The increase in surface hardness can directly improve the coating wear behavior. Wang et al. [14] indicated that the surface wear performance was closely related to the Ti_x_Ni_y_ intermetallic compounds in the diffusion layer, which made the wear mechanism change from adhesive wear to micro-adhesive wear, and the wear resistance was significantly improved. Moreover, Wei et al. [15] revealed that the formation of precipitates resulted in a refined lamellar microstructure with prolonged ageing time, which can furnish the “enveloping effect” and ameliorate the corrosion-resisting property of Ti-Cu alloys. Qin et al. [16] believed that the thermal diffusion could not only form an effective protective film, but that it could also depress pitting growth because of the gradient distribution of Cu and Ni, thus improving the corrosion resistance. Therefore, reasonable diffusion could effectively regulate the surface performance of the coating on TC4 alloy.

Meanwhile, previous studies [11] have revealed that the mechanical property of coating on TC4 alloy is significantly influenced by diffusion temperature, especially for the best performance at 700 °C. To further know the relationship between isothermal diffusion and surface performance such as tribological property and corrosion of the coating on TC4 alloy, the atomic isothermal diffusion behavior and the microstructure of the copper coating on TC4 by the nickel material as the barrier layer after the isothermal diffusion time for 1, 3, 5, and 7 h at 700 °C were studied in this paper. The isothermal diffusion behavior and its effect of microstructure on the tribology and corrosion resistance were also discussed.

## 2. Experiment

### 2.1. Experimental Material

TC4 alloy chemical composition is listed in Table 1. It was processed into a block of the size of 10 × 10 × 5 mm, and the surface of TC4 alloy was ground with 200#, 600#, or 1000# SiC abrasive paper.

The 180 g/L NiSO_4_·6H_2_O, 65 g/L Na_2_SO_4_, 25 g/L MgSO_4_, 25 g/L H_3_BO_3_, and 8 g/L NaCl were selected as the plating solution of Ni plating, and the 210 g/L CuSO_4_·5H_2_O, 60 g/L H_2_SO_4_, and 15 mg/L NaCl were selected as the plating solution of Cu plating. The Ni layer electroplating was carried out by applying 4 V voltage for 6 min. Then, the Cu layer electroplating was carried out by applying 0.65 V voltage for 20 min. The plating process was carried out at room temperature, as diagramed in Figure 1a. The thicknesses of Ni layer and Cu layer were approximately 5 and 9 µm, respectively. The microstructure of Cu plating and Ni plating are shown in Figure 1b. The plating sample was placed in the vacuum furnace (OTF-1200X, MTI-Kejing, Hefei, China) using the vacuum level of 1 × 10^−2^ Pa for isothermal thermal diffusion. The specimens were isothermally diffused for 1, 3, 5, and 7 h at 700 °C, independently.

### 2.2. Microstructural Observation

The cross-section morphologies of the double coatings were characterized by scanning electron microscopy (SEM, Hitachi S4800, Hitachi Limited, Tokyo, Japan) at 20 kV. The element distribution after the samples’ isothermal treatment was confirmed by energy dispersive spectroscopy (EDS). In addition, the worn morphologies of the samples were analyzed, and a laser confocal microscope (LCM, Olympus OLS5000, Olympus Corporation, Tokyo, Japan) was used to show the worn mechanism. X-ray diffraction (XRD, Bruker, Karlsruhe, Germany) technique was used to identify the phase composition of the coating after isothermal diffusion. The XRD experiments were performed in the 2θ range of 20–80° with a step of 0.02°/s.

### 2.3. Tribological and Anti-Corrosion Properties

To explore the influence of isothermal diffusion on the corrosion and tribological property of electroplated Cu/Ni coating, the friction and corrosion tests were performed. The surface microhardness of coating after isothermal diffusion was measured by microhardness tester (HV-1000A Huaying, Laizhou, China). The results were averaged over five tests. The wear resistance of the coating was tested by the ball-on-plate wear tester (MMQ-02G) with the 6 mm GCr15 ball counterpart. The constant load of 3 N was applied normally to the specimen under unlubricated condition at the room temperature. The friction and wear tests were performed by the circular path with a diameter of 3 mm, a rotational speed of 100 r/min, and a sliding distance of 37.70 m.

The samples after isothermal treatment were embedded in the epoxy resin and exposed 1 cm^2^ in contact with the 3.5 wt.% NaCl solution for corrosion test. The computer controlled electrochemical workstation (PARSTAT 2273) was employed in linkage with the traditional corroded batteries, which contained a saturated calomel reference electrode, a platinum (plate) counter electrode, and the coated specimen as the working electrode. The experiment was handled in 3.5 wt.% NaCl solution at room temperature. Prior to electrochemical testing, all specimens were immersed in 3.5 wt.% NaCl solution for 2 h at open circuit potential (OCP) in order to ensure that stationary conditions were reached. Potentiodynamic polarization tests were implemented at sweeping potential between −0.7 and 0.5 V with the sweep rate of 1 mV/s. All of the measurements were performed five times equally to insure good repeatability. The samples were washed with distilled water and dried after corrosion testing.

Tafel extrapolation can be used to obtain the corrosion current density (*I_corr_*) of the coating, that is, the intersection of the Tafel curve anode and the cathode (Figure 2). It can also evaluate the corrosion current density according to the polarization curve to further verify the datum using the Stern–Geary formula [17]:(1)$$I_{corr}=\frac{b_{a}\times b_{c}}{2.303\times R_{p}\left(b_{a}+b_{c}\right)}$$
where *R_p_* is the linear polarization resistance, *b_a_* is the anodic Tafel slope, and *b_c_* is the cathodic Tafel slope.

The *R_p_* was obtained from
(2)$$R_{p}=\frac{dE}{dI}|E=E_{cop}\approx \frac{\Delta E}{\Delta I}$$
where Δ*I* is the current density in the unit of A/cm^2^, and Δ*E* is the potential in volts.

## 3. Results and Discussion

### 3.1. The Phase Transitions and Microstructure of the Coating

Isothermal treatment is an atomic diffusion process that could lead to a variety of microstructures of the coating [18]. The XRD results showed that solid solution of α(Cu, Ni)—Cu_0.81_Ni_0.19_ was formed after isothermal diffusion for 1 h at the Cu/Ni interface (Figure 3a). Its relative position is labelled at region A in Figure 3b. As the isothermal diffusion time increased, the Ni_3_Ti, NiTi_2_, and NiTi phases gradually formed within the Ni/Ti interface (labelled at region B in Figure 3e), which can be expressed as
(3)$$3Ni+Ti=Ni_{3}Ti,$$
(4)$$Ni+2Ti=NiTi_{2},$$
(5)$$Ni_{3}Ti+2NiTi_{2}=5NiTi.$$

The formation of Ni_3_Ti in the Ni side of the Ni/Ti interface and NiTi_2_ in the Ti side increased boundary diffusion barriers and further inhibited the NiTi formation process. The smaller Gibbs free energy changes for the reaction between $Ni_{3}Ti$ and $NiTi_{2}$ (Equation (5)) compared to Ni+Ti=NiTi reaction implied the low possibility of the formation of NiTi [19]. In addition, as the isothermal diffusion time increased, Ti atoms with strong diffusion ability formed the CuTi_2_ and CuTi phases on the Cu side through the intermediate Ni layer.

It is worthy to note that from Figure 3c the numerous continuous microvoids or cracks were observed at the Ni side of the Ni/Ti interface after isothermal diffusion for 5 and 7 h. This was due to the fact that phase transitions from Ni_3_Ti and NiTi_2_ to NiTi will inevitably lead to the change of volume, which will result in stress increase that induces the formation of voids or cracking of nearby phases interfaces during the diffusion process [18]. With the extension of diffusion time, the number of microvoids or cracks increased in the diffusion layer (Figure 3c,f). These results indicated the occurrence of the Kirkendall effect in Ni/Ti diffusion couple during diffusion. It is well known that an atom can move to a vacant lattice site and exchange atomic and vacant positions, and that the diffusion taken place in an interface can cause atomic flux and vacancy flux in the opposite two directions [20]. In this study, the net diffusion of atoms could move from the Ni layer with higher diffusion coefficient to the Ti layer with lower diffusion coefficient [21], resulting in Kirkendall voids. Under such a circumstance, the vacancies would diffuse from the Ti layer toward the Ni layer. With the extension of diffusion time, the growth of Kirkendall voids accelerate, and they start to connect to each other and merge into larger microvoids defects within the diffusion layer [12]. Additionally, the flocculent Kirkendall diffusion channel can be found on the inner side of the Ti substrate, which is due to the rapid diffusion of Ni atoms to Ti substrate and vast vacancies migration [22,23]. Apparently, the increased density of the microvoids or cracks and Kirkendall diffusion channel made the microstructure of the diffusion layer imperfect with the extension of diffusion time, as schematically shown in Figure 3d.

### 3.2. Effect of Isothermal Treatment on Diffusion Behavior

Figure 4 displays the element distribution and microstructure of the specimens after isothermal diffusion. Obviously, the interface in the Cu/Ni/Ti layers gradually became blurred and disappeared with the extension of diffusion time, indicating that the diffusion behavior of the atoms increased significantly. In addition, the diffusion of the Ni atoms was the most active. It was mainly because a large number of dislocations and point defects were generated during the copper plating process, which caused the internal grain boundaries to be in a high energy state [24]. The Ni atoms can form a rapid diffusion channel along the copper grain boundary with a faster diffusion rate [25,26]. The thickness of the diffusion layer can accurately reflect the diffusion level of one material, which is definitely affected by the diffusion time [27]. At a certain temperature, the dependence of the thickness of the diffusion layer on the diffusion time can be described by the following empirical formula [28,29]:(6)$$w^{2}=Kt$$
where w is the thickness of diffusion layer, *K* is the correlation coefficient, and *t* is the diffusion time.

Figure 5 depicts the change of the squared thickness when the isothermal diffusion time increased. It can be shown that the thickness of the Cu/Ni/Ti diffusion layer increased with diffusion time, but that the slope (*K*) decreased gradually. The decreasing *K* signified that the atom diffusion became more difficult during the isothermal diffusion process. It is well known that the diffusion rates of Cu, Ni, and Ti atoms are constant under the isothermal diffusion condition of 700 °C, which means that the atoms were hindered in the process of isothermal diffusion. Chen et al. [30] and Liu et al. [31] found that atom diffusion ability is related to atom concentration, and the formation of multiple reaction layers in interface also has a certain negative effect on atomic diffusion. Meanwhile, Tavoosi [12] suggested that the formed microvoids or cracks could act as the barrier in double-diffusion system, owing to the blocking of diffusion paths. Therefore, the thickness of isothermal diffusion layer is synthetically affected by concentration of diffusion atoms, the Ni_x_Ti_y_ and Cu_x_Ti_y_ intermetallic compounds layers, and the presence of microvoids or cracks when the diffusion time increases. The longer the isothermal diffusion time, the more obvious this inhibition effect becomes.

### 3.3. Effect of Isothermal Diffusion Behavior on Tribological and Anti-corrosion Properties

Previous studies [32,33] on the nickel or copper layer on the surface of TC4 alloys suggest that the varieties of the diffusion layer microstructure could substantially affect the mechanical properties of coatings. The effect of diffusion time in the microhardness and tribological property of the coating heat-treated at 700 °C are showed in Figure 6. It is explicitly shown that the hardness of the coating was significantly improved, which we regarded as resulting from the solid solution strengthening (α(Cu, Ni)) and precipitation strengthening (Cu_x_Ti_y_, Ni_x_Ti_y_) in the diffusion layer [19]. Wang et al. [14] showed that the dispersion of the hard phases in coating can effectively enhance the resistance of coating to high contact stress, meaning that the increased surface hardness can directly optimize the coating wear behavior. As compared with untreated samples, the heat-treated samples exhibited lower wear mass loss and smaller friction coefficient, especially at the isothermal diffusion for 3 h (Figure 6a,b). Obviously, when the diffusion time was 3 h (Figure 6c), the wear marks were flat and smooth, accompanied by apparent plastic deformation, and demonstrating a micro-adhesive wear mechanism. The Cu layer had a good ductility, which can effectively boost the brittle fracture resistance of the coating. In the process of wear and tear, the Cu layer with lower hardness first underwent a certain degree of deformation and the contact areas of adhesion wear increased gradually. On the other hand, higher hardness phases can oppose more tear loads and restrict soft Cu layer from further deformation. In this case, the Cu layer can significantly alleviate the crack initiation and propagation, owing to its excellent ductility. When the diffusion time was increased to 7 h (Figure 6d), the coating layer was gradually hardened due to the formation of vast intermetallic compounds (Cu_x_Ti_y_, Ni_x_Ti_y_) and solid solution (α(Cu, Ni)) at the surface, accompanied by numerous microvoids and cracks caused by the Kirkendall effect. Due to the increased internal stress, the coating layer was easy to break and the hard phases were inclined to fall off. Furthermore, deep furrows and serious cracks were observed, which exhibited poor wear resistance. It was quite clear that the friction mechanism transformed into abrasive wear. Therefore, reasonable control diffusion time could make the hard phases and the soft Cu layer coordinate and cooperate with each other to significantly improve the wear performance of coating.

Figure 7 shows the polarization curves of the coatings after isothermal diffusion treatment for 700 °C. The corrosion parameters, such as the corrosion current density (*I_corr_*) and corrosion potential (*E_corr_*), could be attained by Tafel extrapolation [8,17] (as listed in Table 2). Compared with the *I_corr_* of the TC4 substrate, the *I_corr_* of the coatings decreased remarkably. Evidently, with the increasing heat treatment from 1 to 3 h, the *E_corr_* increased from −354.07 to −201.14 mV, and the *I_corr_* decreased from 6.04 × 10^−3^ mA/cm^2^ to 0.514 × 10^−3^ mA/cm^2^, meaning enhanced corrosion resistance. This can be explained from the perspective that the formation of Cu_x_Ti_y_ precipitates owing to the Ti atoms approaching the Ni/Cu interface led to the thin slice multilayer microstructure. It can provide the “enveloping effect” and effectively block expansion of corrosion [15]. However, the *E_corr_* decreased to −276.62 mV, and the *I_corr_* increased to 2.68 × 10^−3^ mA/cm^2^ after the isothermal diffusion increased to 7 h, indicating a degraded corrosion resistance. This was closely related to the microstructure of the diffusion layer. With the extension of the diffusion time, the Kirkendall effect was even more pronounced. A large number of Kirkendall voids and Kirkendall diffusion channels could exist to destroy the integrity and continuity of the multilayer microstructure, losing its inhibiting effect on corrosion and even accelerating the corrosion of the coatings [34]. Therefore, reasonable control diffusion time can obtain good Cu/Ni corrosion-resistant coatings on TC4 alloy.

## 4. Conclusions

In this work, the Cu/Ni double coatings on TC4 alloy were prepared by the diffusion bonding under isothermal treatment at 700 °C for 1, 3, 5, and 7 h. The microstructure diffusion behavior, tribological property, and corrosion resistance were investigated. With the increase in diffusion time, the occurrence of Kirkendall diffusion was observed, and the Kirkendall diffusion channel appeared at the Ti side in the Ti/Ni interface. Meanwhile, numerous continuous microvoids or cracks were formed at the Ni side in the Ti/Ni interface. The thickness of the diffusion layer increased nonlinearly due to the inhibition of the diffusion layers, which significantly affected the behavior of isothermal diffusion of Cu, Ni, and Ti atoms. The hard phases (Cu_x_Ti_y_, Ni_x_Ti_y_, α(Cu, Ni)) and the soft Cu layer were found to coordinate and cooperate with each other to significantly enhance the wear resistance. Corrosion resistance of coating was found to increase firstly and then decrease with the increasing of isothermal diffusion time, which was related to the transition from refined lamellar multilayer to microstructure of continuous microvoids or cracks. In all, this work showed that the surface performance, such as the wear resistance and corrosion resistance of coating, can be significantly increased because of formation of continuous refined lamellar multilayer and microstructure by using appropriate diffusion time (such as 3 h).

## Figures and Tables

**Figure 1 materials-12-03884-f001:**
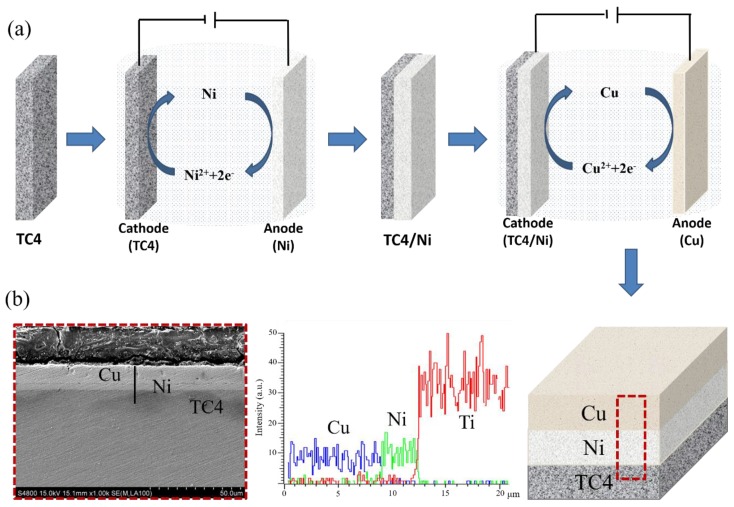
The electroplated Cu/Ni coating on TC4 alloy: (**a**) the reaction test process; (**b**) the microstructure and the corresponding element concentration.

**Figure 2 materials-12-03884-f002:**
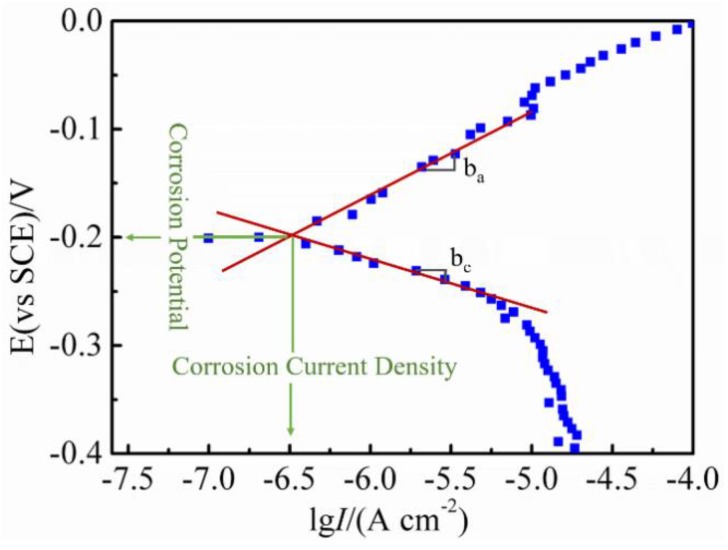
Estimation of corrosion current density (*I_corr_*) from the polarization curve.

**Figure 3 materials-12-03884-f003:**
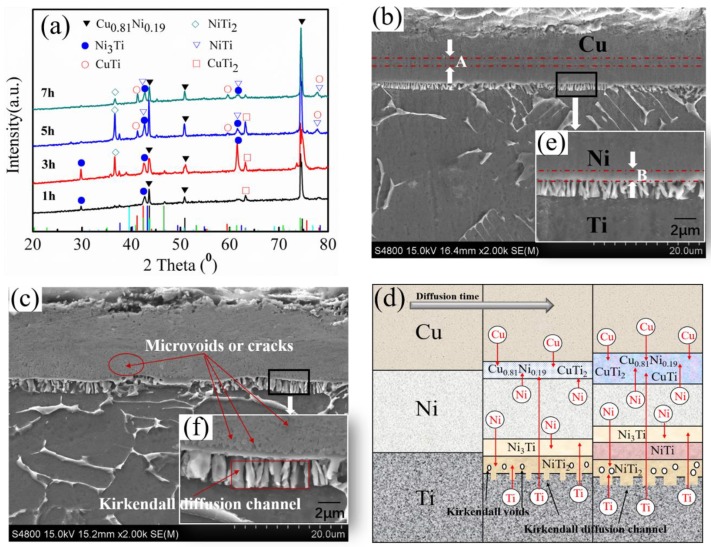
The electroplated samples’ isothermal treatment at different times at 700 °C: (**a**) surface XRD images; (**b**,**c**) cross-sectional microstructures under 1 and 7 h isothermal diffusion; (**d**) atom diffusion reaction processes; (**e**,**f**) the high magnification microstructure of the Ni/Ti interface. As represented, the α(Cu, Ni) solid solution layer and an obvious diffusion layer was observed in the Cu/Ni interface. B represents the Ni_x_Ti_y_ intermetallic compounds layer and a continuous reaction layer along the Ni layer and the Ti substrate; (**c**,**f**) shows the area with microvoids or cracks and the Kirkendall diffusion channel.

**Figure 4 materials-12-03884-f004:**
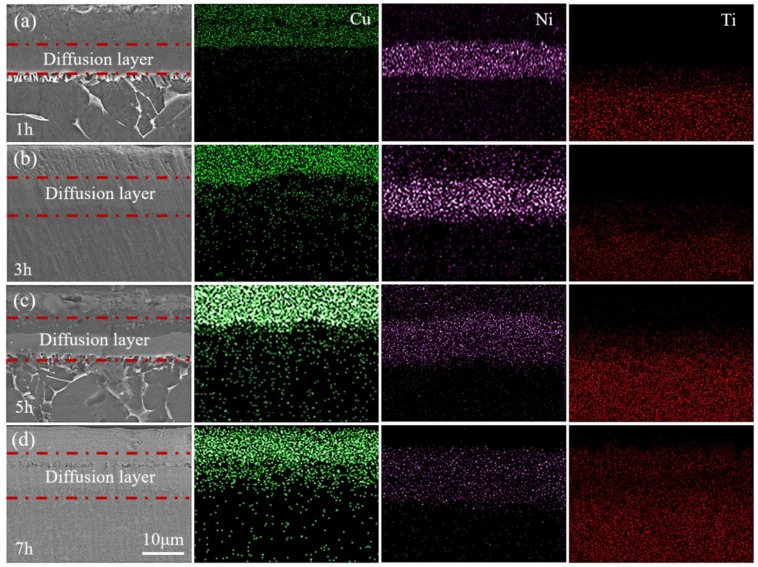
The microstructures and elements distribution of the samples after isothermal treatment at different times at 700 °C: (**a**) 1 h, (**b**) 3 h, (**c**) 5 h, and (**d**) 7 h.

**Figure 5 materials-12-03884-f005:**
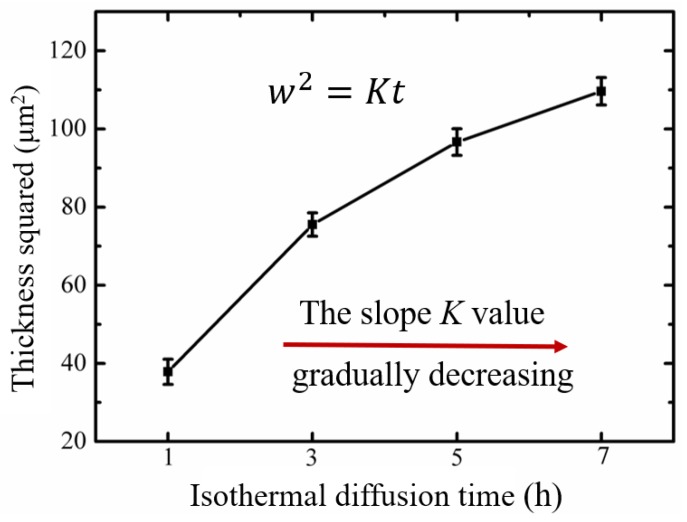
Isothermal diffusion time dependence of the thickness of diffusion layer.

**Figure 6 materials-12-03884-f006:**
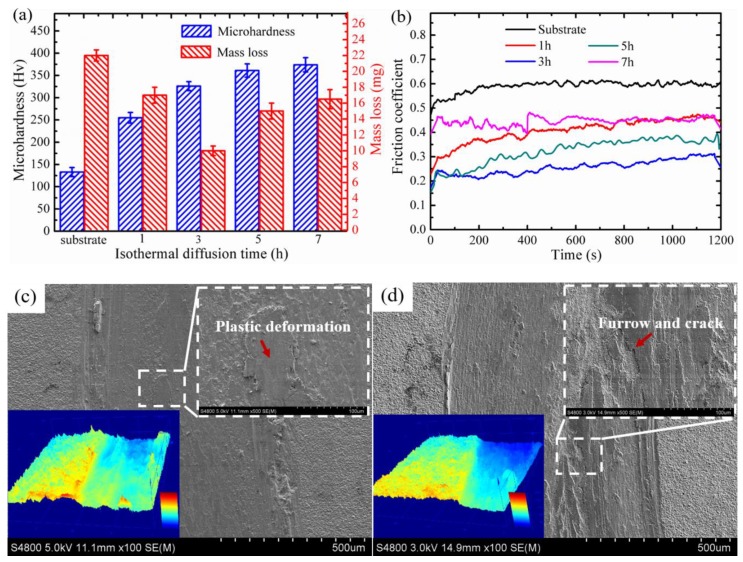
The electroplated samples’ isothermal treatment at different times at 700 °C: (**a**) surface microhardness and mass loss; (**b**) friction coefficients curves. (**c**,**d**) show the worn surface SEM and laser confocal morphologies of heat treatment for 3 and 7 h, respectively.

**Figure 7 materials-12-03884-f007:**
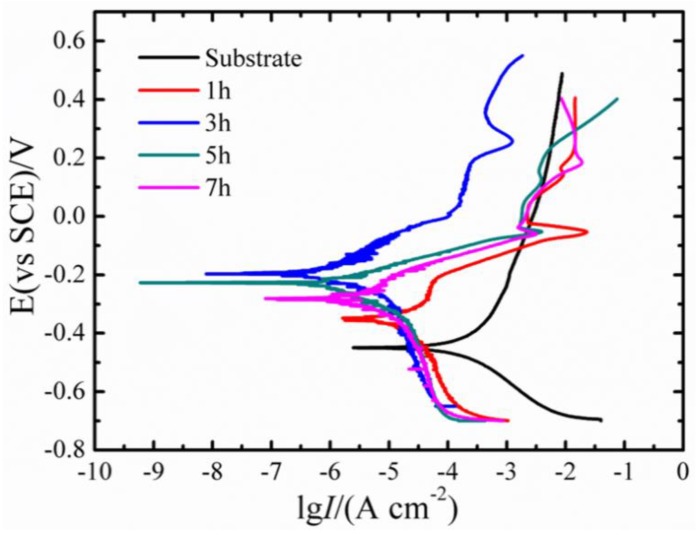
The polarization curves after isothermal diffusion treatment for 700 °C.

**Table 1 materials-12-03884-t001:** Chemical compositions of the TC4 alloy (wt.%) [11].

Elements	V	Al	Fe	Mn	Zn	Si	Ti
Nominal	3.3–4.5	5.0–6.5	0.3–0.9	0.5	0.3	0.4	Bal

**Table 2 materials-12-03884-t002:** Electrochemical data for the coating from potentiodynamic polarization.

Diffusion Time (h)	*E*_corr_ (mV.SCE)	*I*_corr_ (mA/cm^2^)	*b*_a_ (mV/dec)	*b*_c_ (mV/dec)
Substrate	−455.42	11.17 × 10^−3^	132.48	−63.72
1	−354.07	6.04 × 10^−3^	99.19	−125.39
3	−201.14	0.514 × 10^−3^	96.98	−59.43
5	−231.72	0.705 × 10^−3^	52.90	−73.90
7	−276.62	2.68 × 10^−3^	101.51	−121.39

Note: *E_corr_* and *I_corr_* are the corrosion potential and corrosion current density, respectively; *b_a_* and *b_c_* are the anodic and cathodic Tafel slopes, respectively.

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
