# Peer review of "Isothermal Diffusion Behavior and Surface Performance of Cu/Ni Coating on TC4 Alloy"

_materials, 2019, doi:10.3390/ma12233884_

Round 1

Reviewer 1 Report

1.       In Fig.1b the units are needed on the y-axis. Maybe readers propose that this is a concentration in wt. or at. percent, but it looks as counts or arb. units.

2.       At line 123 please replace Cu0.81Ni0.91 to Cu0.81Ni0.19.

3.     In Fig.3a the peaks of NiTi phase for sample isothermally treated for 7h is the same as for Ni3Ti phase peaks for samples isothermally treated for 1, 3 and 5 hours. In scheme (Fig. 3d) the CuTi2 and CuTi phases formed after long isothermal treatment, but according to XRD, the CuTi phase is formed after 1h isothermal treatment. As opposite in scheme, NiTi2 is formed before Cu-Ti phases, but in accordance with XRD, this phase found after 3h isothermal treatment. Please check the XRD patterns and scheme in Fig. 3d.

4.       It is difficult to see microvoids in Fig.3f. In fig. 4d the microvoids are  visible better.

5.       Please use the same font for w in eq. (6) and line 168.

6.       How the boundaries of the diffusion layer (horizontal red lines in Fig. 4a) was found? I can't see any connection of diffusion boundary lines with EDS maps shown in Fig. 4b and Fig. 4c. 

7.       Please add units on the x-axis in Fig.5.

8.     The elements on EDS spectra are wrong detected. For example at spectra for 7h isothermal treatment sample the first peak denoted as Cl is O peak. Please check this and other spectra.

9.     It can be seen form Fig.7b that the corrosion products composition is different for samples. For example, for the sample with highest corrosion resistance the corrosion products mainly consist of titanium oxide, but for others is mostly Cu,Ni oxide. Maybe it is better to say about this fact in the article text. 

Reviewer 2 Report

The topic of the manuscript “ Isothermal diffusion behavior and surface

3 performance of Cu/Ni coating on TC4 alloy” is seems to be interesting and important. According to my opinion this study is a valuable work, the paper is well written, the research is well designed and decisions are justified, (I have some doubts about the paper which I listed below), therefore I suggest this paper for “minor revision”:

Minor problems:

I do not like the phrase “composite layer”, because I think it is not the fact now. The authors did a double layer from Cu and Ni. They can call the Ni as barrier layer, but the whole structure is not a composite layer. 4 suggests that the diffusion of Ni was the most active. It is a bit strange for me, since I would say that the Cu should have been. Please explain “this anomaly” in the paper. The quality of Fig. 7b is very bad. Are these the salts on the surface? Please improve it or delete. I do not like the following sentences in the Conclusions: ”It is found that multiple diffusion layers are formed in the Cu/Ni and Ni/Ti interface, and some new phases of Ni3Ti, NiTi2, NiTi, CuTi and CuTi2 are found in the composite coating after isothermal diffusion. It is shown that the isothermal diffusion behavior of Cu, Ni and Ti atoms has been significantly influenced by the diffusion time.” These are evidences, I mean you waited exactly these results. Please highlight the real conclusions of your work.
